# A Review Regarding the Use of Molasses in Animal Nutrition

**DOI:** 10.3390/ani11010115

**Published:** 2021-01-07

**Authors:** Attilio Luigi Mordenti, Elisa Giaretta, Luca Campidonico, Paola Parazza, Andrea Formigoni

**Affiliations:** 1Dipartimento di Scienze Mediche Veterinarie (DIMEVET), Alma Mater Studiorum—Università di Bologna, Via Tolara di sopra 50, 40064 Ozzano Emilia, Italy; luca.campidonico@unibo.it (L.C.); paola.parazza@unibo.it (P.P.); andrea.formigoni@unibo.it (A.F.); 2Dipartimento di Biomedicina Comparata ed Alimentazione (BCA), Università di Padova, Via dell’Università 6, 35020 Legnaro, Italy; elisa.giaretta@unipd.it

**Keywords:** sugar cane molasses, sugar beet molasses, animal nutrition, animal production, circular economy

## Abstract

**Simple Summary:**

The aim of the authors is to make a summary of the possible applications of molasses in animal nutrition, how to improve hays and silage qualities for beef and dairy cattle; to enhance industrial byproducts values by liquid feed in swine production; and to improve with feed blocks the extensive livestock production efficiency (cows, buffaloes, sheep, goats and pigs). Focus is both on characteristics feed based on molasses and on ruminal fermentation: the techniques of production, conservation and administration to animals have been widely described as being capable of positively influencing animal performance, milk and meat quality, as well as animal welfare.

**Abstract:**

In the past fifty years, agriculture, and particularly livestock production, has become more resource-intensive, with negative implications regarding world environmental status. Currently, the circular economy 3R principles (to reduce, reuse and recycle materials) can offer many opportunities for the agri-food industry to become more resource-efficient. The closed-loop agri-food supply chain has the great potential of reducing environmental and economic costs, which result from food waste disposal. To meet these principles, the use of crop byproducts, such as molasses, in animal nutrition improves the nutritive value of coarse and poorly desired feedstuff, which could present a real opportunity. The aims of this study were to summarize the possible applications of molasses for animal nutrition, to improve hay and silage quality for beef and dairy cattle, to enhance industrial byproduct values using liquid feed in swine production, and to improve extensive livestock production with feed blocks. The study focused on both feed characteristics, based on molasses, and on ruminal fermentation of its carbohydrates; the techniques of the production, conservation and administration of molasses to animals have been widely described as being capable of positively influencing animal performance, milk and meat quality.

## 1. Introduction

Molasses is a thick brown-colored liquid with a syrupy consistency, which is the residue remaining after sugar extraction when it is no longer possible to conveniently obtain sucrose from the latter for simple crystallization [1]. However, this definition is not sufficient to characterize molasses; in fact, in sugar beets (*Beta vulgaris*), the sucrose is extracted from the deep root while, in sugar cane (*Saccharum officinarum*), it derives from the trunk medullary tissues. The byproducts resulting from these processes are certainly similar, but they do present different organoleptic and nutritional characteristics that are not negligible or estimable.

The data highlight the fact that global sugar production as shown by OECD/FAO [2] increased steadily during the period from 2016 to 2018, and that Brazil, India and Thailand were the most important producers of sugarcane (33, 28 and 12 Mt of sugarcane produced, respectively). Africa (11 Mt), China (10 Mt) and the United States (7 Mt) were placed in the middle position (sugar production being considered both that of cane and of beet). The European Union maintains its position as the world’s largest producer of sugar beets (17 Mt). Since 2006, sugar beet production in Italy has dropped drastically, from 8.6% to 3.9% of the European Union (EU) production due to the EU reform, which liberalized the sugar market in all of Europe, and to the lower production costs of sugar cane. Molasses derived from sugar cane, which comes mainly from South America, is the most important for zootechnical use.

## 2. Molasses, Characteristics and Uses

Regardless of its origin, molasses is considered an energetic feedstuff due to its high content of easily fermentable sugars. However, it is also rich in mineral salts, present in a bioavailable form. Low cost allows it to be a very popular feedstuff as a partial substitute for cereals (it has approximately 2/3 of the energy value of corn) in feed formulations and in rations for many species of zootechnical interest. As regards some physical-chemical properties and total energetic value, beet and cane molasses have quite similar characteristics. In both cases, they are liquids with a high-density (1,3) and an acidic pH (5 to 5.8). However, there are some significant differences between the two products, particularly regarding sugar composition, nitrogenous fraction and mineral salts (Table 1, Table 2 and Table 3).

Concerning the total sugar amount, it ranges from approximately 48 to 53% for both types of molasses. These data are not negligible because, in addition to the increased product quality, together with mineral salts, sugar has an antimicrobial action due to its high osmotic potential, which influences its stability and shelf life. In this sense, safety must also take into account the problem of mycotoxins, which often contaminate many feedstuffs, but only partially molasses [3]. Upon arrival to the user industry, molasses has slightly different chemical characteristics as compared to those which it should have from a commercial point of view. The product is standardized; the sugar content reaches a high level of more than 48% and has a water content lower than 25%. The aim is to keep the conservation unchanged and the product quality constant. Another parameter usually used by the sugar industry is the Brix grade, which allows measuring dry matter and sucrose content simultaneously using a refractometer. Reference guideline values varied from 78 to 85° Brix. Sucrose content differs between beet and cane molasses (60.9 ± 4.4 vs. 48.8 ± 6.4% of DM, respectively) [4]. However, to be more specific regarding sugar content, the parameter which differentiates the different nature of molasses, the inverted sugar amount, should be noted. These data are calculated using the total sugars as inverted (TSAI) index. The inverted sugars are monosaccharides (fructose, glucose), which have a reduction reaction with Fehling’s solution and sugars which do not ferment because they are bound to amino acid nitrogen (Maillard reaction) while maintaining reduction properties. The amount of reducing sugars in beet molasses is from approximately 0.5 to 1.5%, while, in cane molasses, it is from 10 to 20%, with an average fructose/glucose ratio equal to 1.58. The increased content of reducing sugars (non-fermentable and not biologically available) [5] in sugarcane molasses, which can increase to 3–5% as compared with sugar beet molasses, is explainable due to the fact that the reducing sugars are more represented in the trunk of the sugarcane plant, and also have the lowest quality extraction processes used in the producing countries [6].

As regards crude proteins, the amount is evaluated by nitrogen content (N × 6.25). Beet molasses content is higher (8.7% as feed) than that of cane molasses (3.7% as feed). In Italy, elevated beet molasses data could be related to the high concentration of Northern Italian clay soils, a result of fertilization. For these reasons, the molasses crude protein content is, for the most part, composed of nitrates (nitrogen of mineral origin) as compared to soluble nitrogen (of organic origin), which is composed of free amino acids (in particular, glutamic acid, aspartic acid, leucine, tyrosine, arginine and histidine; Table 4) and by nitrogen bases (betaine, guanine, xanthine). Given the low value of these nitrogenous fractions (amino acids and betaine apart) and their modest representation in molasses, they are of limited importance. In any case, the important biological role (methyl donor) exerted by betaine (0.5–0.7%) should be pointed out; like methionine, it can contribute to reducing the fatty liver problems so recurrent in dairy cows [7,8,9,10,11,12].

Considering the vitamin content of molasses shown in Table 5, it can be stated that it is poor in fat-soluble vitamins [1]. As regards the water-soluble vitamin, in particular, group B, both types of molasses have a low concentration in thiamine. Vitamin B1 deficiency, which plays a particularly important role in glucose metabolism, requires an adequate correction in the case where a large amount of molasses is used in the rations of monogastric animals [13]. In polygastric animals, under normal conditions, this vitamin is synthesized in sufficient quantities by the ruminal bacterial microflora. Despite the vitamin deficiencies mentioned above, the presence of considerable amounts of biotin in cane molasses (Table 5), which is also present in a totally bioavailable form, should be noted. Biotin content takes on particular significance not only in pig feed (where vitamin H deficiencies are common and cause foot damage), meat quality (fat) and fertility but also in high-productive dairy cows, where it appears to have interesting effects both in foot pathology prevention and in milk production and quality [14]. In particular, in 2011, Chen published [15] a meta-analysis to verify the efficacy of biotin supplementation in dairy cattle. The authors analyzed eleven papers in which it was shown that supplementing lactating cows with 20 mg per day of biotin could increase milk production by 1–2.9 kg/head/day, probably as a consequence of increased ingestion due to an improvement in the rumen digestion of fiber. In the same year, Lean and Rabiee published [16] a review of randomized controlled clinical trials. The authors included nine scientific papers, six of which had the aim of verifying the possible positive or negative effects of biotin (20 mg/head/day) on milk production and another three which studied its impact on the hoof health of dairy cows. In addition, in this case, an increase in production was ascertained; however, no effect on milk quality (fat and protein) was mentioned.

As regards the mineral fraction present in the ash content, it constitutes 8–9% of the molasses with very similar values in both beet and cane molasses (Table 1). However, a higher proportion of calcium and phosphorus ions can be noted in cane molasses, while sodium and potassium ions prevail in beet molasses, probably due to the specific land allocations (Table 3). This must be taken into account in feed formulation in order to keep the mineral balance of the ration and, in particular, the low content of both calcium and phosphorus correct. The drawbacks attributed to molasses mainly concern its potassium salt content, which can constitute up to 1/3 of the ash content. Potassium salts have demineralizing and laxative effects [17], but, with their rational use, keeping in mind the balance between potassium and sodium in ration formulation, it is possible to overcome the above-mentioned drawbacks and, at the same time, take advantage of the favorable properties of this product.

A particular consideration had to be made regarding the amount of sulfur in the molasses, as shown in Table 3. It is interesting to note that the total sulfur content for beet molasses (5.6 g/kg DM; CVB Feed Table data) and the sulfur amount of cane molasses, for the most part, represented by sulfates in an inorganic form (8.2 g/kg DM; Feedipedia data). The EFSA opinion [18] provides comprehensive literature about sulfur and its administration to the animals. In numerous forages, feed materials and feed additives, the sulfur amount may be relevant. Kamphues’ review [19] provides quantitative information regarding the current levels of sulfur and sulfate in various feedstuffs and also in drinking water. Excessive consumption of sulfur may have multiple effects on animal health and/or production; much research has been carried out to define a clear dose-response effect of sulfur and sulfates, both in ruminants [20,21] and in pigs [19,22]. The National Research Council (NRC) [16] data set the maximum tolerable level (MTL) of sulfates for ruminants at <3 g S/kg DM in diets rich in cereals and <5 g S/kg DM in diets rich in roughage. For non-ruminants (pigs and poultry), the MTL is set at <4 g S/kg DM [16]. As regards drinking water, the MTL for sulfate (NRC) is 600 mg/L in high concentrate diets and 2500 mg/L in high forage diets, while Kamphues recommended 500–800 mg/L [19]. For non-ruminants (pigs and poultry), the MTL of the NRC is set at 3000 mg/L for pigs, while Kamphues proposed 1600–1800 mg/L for pigs.

A recent study [4] showed that the organic acid composition differed among types of molasses. Lactic acid was more concentrated in cane molasses as compared to beet molasses (4.69 vs. 3.48 as feed), varying from 9.77% maximum to 1.23% minimum in cane molasses. Aconitic acid, synthesized by the dehydration of citric acid using sulfuric acid, was found only in cane molasses, while glycolic acid was found only in beet molasses. The total sum of organic acids ranged from 2 to 14% as feed. Sulfates, phosphates and chlorides had a higher concentration in cane molasses, which showed a lower cation/anion balance (DCAD) as compared to beet molasses (4.47 vs. 53.94 meq/100 g). In the cane typologies, it varied from +117.63 to −58.59 meq/100 g while, in the beet typologies, it varied from +129.20 to +3.24 meq/100 g. If one is concerned about using molasses in dry-fed cows because of the high variability of potassium content, a liquid feed with low, neutral or even anionic DCAD could fit perfectly for this group of cows. On the contrary, nutritionists could choose higher DCAD liquid feeds to be used during the summer [23].

The physical characteristics of molasses make it a very particular feedstuff with a large number of possible applications in both feed mills and farms. The viscosity of both types of molasses varies significantly in relation to both the production area and the ambient temperature. In some cases, it may create problems that are not easy to resolve. Viscosity is generally influenced by dry matter and polysaccharide content and by ambient temperature. In any case, the viscosity of cane molasses is much higher and has a wider range of variation than that of beet molasses, the range of which is significantly lower. For example, at a temperature of 20 °C and 75% dry matter, the viscosity of cane molasses varies from 14,000 to 7000 centipoises, while that of beet molasses ranges from 2000 to 1400 centipoises. With an additional increase in temperature, there is a decrease in viscosity; however, the data variation range of cane molasses is always greater than that of beet molasses.

The most important organoleptic characteristic of molasses is represented by its sweetening property and its taste, which results in being particularly appreciated by the animal (appetizing role) and which increases the ingestion of feedstuffs that do not always have good palatability. Another technological characteristic of molasses is represented by its binding properties: they are exploited by the feed industry during the preparation of compound feed through the addition of molasses (cold or hot) and pelleting techniques. The binding and appetizing properties of both types of molasses have recently been recognized by the latest regulation of Parmigiano-Reggiano PDO cheese, which also allows its use in the preparation of supplements for lactating cows [24,25]. Another non-negligible particularity of both types of molasses is their ability to reduce powder content, a frequent cause of irritation of the first respiratory tract of both animals (pigs above all) and workers.

## 3. Molasses in Liquid Feed

Traditionally, the use of dry feed for animals has always been a necessity imposed by their better preservability over time and by easy transport and distribution. For these reasons, for a long time, a series of liquid byproducts of industrial derivation remained excluded, which, instead, could have had their correct and profitable position in the zootechnical field. This fact led both to the inadequate exploitation of some profitable resources with consequent economic losses and to enormous ecological damage caused by the reckless discharge of these substances, having a high biological oxygen demand (BOD), potentially very polluting into watercourses. With this in mind, interest in the rational exploitation study of the byproducts of the food industry has increased in order to enhance the specific nutritional values of molasses.

The use of sugar cane molasses in animal nutrition is certainly much higher in industrialized countries than in tropical ones. This is due both to the different characteristics of local zootechnical farms and also because, in industrialized countries, there has been a sudden development of companies, the aim of which is the proper disposal of byproducts derived from the food industry. It is useful to point out the notable ecological value of this type of activity [26].

Liquid feeds based on cane molasses can improve ration palatability and reduce any sorting activity of the animal [27]. Moreover, when a dry total mixed ration (TMR) is prepared without silage, liquid feeds are very efficient in reducing the presence of dust. As was correctly pointed out by Feroci and Nistri [28], it is important to examine the many practical problems which can be resolved on many breeding farms by using an appropriate liquid diet: (a) enhancement of the industrial value of byproducts; (b) savings on farm feed costs; (c) better organization of work schedules and (d) reduction in powders with resulting technological and health benefits.

The formulation of a liquid feed can include all those byproducts of different natures and origins, which can be used in the diets of different animal species of zootechnical interest, obviously in compliance with the current legislation in the area. Liquid feed has a low cost due to the nature of the raw materials and, in particular, production technology. In fact, being marketed in this form, they do not must be subjected to drying costs, which often worsen their quality. There is no doubt that the disadvantages of the byproducts, in which water is the main component, are linked to conservation and storage and transport costs. Here is how the use of molasses can be useful. Molasses is the main component of liquid feed, contributing 60% (and more) to its composition due to its preservative properties, the specific osmotic power of sugars and salts, which also reduce its mycotic contamination, and the possible consequent problems of frequent and alarming mycotoxicosis in today’s animal production. Moreover, the mycotoxin contamination of molasses has already been described [3,29]. The remaining portion of a liquid feed is made up of raw materials of different natures, the characteristics of which change according to the production processes from which they derive. Due to their different dietetic and nutritional characteristics, these other byproducts have a different percentage of inclusion in feed, depending on the animal they are destined for and its particular physiological moment. A brief overview of the raw materials which can commonly be included in the formulation of liquid feeds could also be useful at this point. The most widely used byproducts are, (a) the yeast extract (derived from the production of vegetable broth preparations), (b) the soluble corn distillers (derived from glucose production), (c) fruit syrups, (d) glycerol (from glucose fermentation), (e) concentrated whey, (f) protein derivatives of lysine processing, (g) calcium lignosulfonate (a byproduct of wood pulp), (h) beet protein concentrate (CPB) represented by the distiller of beet molasses with high nitrogen and mineral contents which derive from alcoholic fermentation, (i) beet distillers (which have undergone a process of demineralization), (j) animal fat (usually a mixture of tallow and lard in variable proportions) or vegetable fat with the addition of emulsifiers and antioxidants and (k) yeasts. These byproducts are supported by other raw materials of a liquid nature, normally present in a feed mill, which is not industrial byproducts: propylene glycol, glycerol, liquid urea, soybean oil, amino acids, mineral and vitamin supplements, lysine, methionine, propionate, orthophosphoric acid, citric acid, and other organic and inorganic acids.

In high producing cows, the digestion of the different nutrients in liquid feeds changes depending on their source; in particular, digestion rates (Kd) vary in relation to the origin of these nutrients, while passage rates (Kp) are generally the same for all the components solubilized. The passage rate for liquid feeds is much higher than for solid feeds. In high producing cows with high dry matter intake, a Kp of 1.5–2.0%/h is estimated for forage, 5–8%/h for concentrates, and more than 14–16%/h for nutrients solubilized in rumen liquor (Formigoni, data not published). According to these data, a reasonable amount of nutrients from liquid feeds would reach the intestine and could be absorbed depending on the respective digestion rates.

On farms, the molasses liquid feed is stored in tanks of different capacities depending on use. The tanks are usually equipped with a pump for sampling and a flag, with a sprayer bar, for unloading. On pig farms, which usually use fatty liquid feed, the tanks are predisposed with agitators inside in order to keep the product homogeneous. There are different possibilities for administering a liquid feed. First, it is possible to spray and coarse mix the fodder directly into the feeder if sufficient manpower is available on the farm, or it can be administered by self-feeders. However, the best results are obtained by its addition, in a gradual and uniform way, to a unifeed mixer wagon with minimal time loss and good quality of the TMR obtained, which is homogeneous and free of powder for animals and workers. Furthermore, it is interesting to add liquid feed to the forage during ensiling when it is minced; this use, in addition to increasing the nutritional value of the ensiled mass, improves its shelf life, both enhancing lactic fermentation and promoting pH drop, and, due to that improving the silage outcome.

To conclude, it can be stated that cane molasses has assumed a first-rate role in animal feed, and its use is a common practice already well known and particularly effective. In fact, it has won the confidence of farmers because (a) it allows increasing dry matter intake in ruminant diets and reducing cow sorting activity, thanks to its high palatability; (b) it enhance the energy part of the TMR without interfering with the fiber, to obtain easily digestible sugars, and (c) it balances the presence of soluble or easily degradable nitrogenous sources and enhances low-quality fodder and cereal grains [27].

The economic convenience of using liquid feed is usually based on comparison with cereal grain prices (corn in particular). However, this comparison is not correct since it refers to a feed and not to raw material. In fact, the nutritional value of a liquid feed is improved by the addition of supplements, such as fats, mineral-vitamin supplements and nitrogenous sources, which improve and complete the final quality. It should also be noted that all raw materials were used in a liquid state, including molasses, present the various nutritive principles in a more bioavailable form as compared to the traditional raw materials that normally are used in flour or pellets. One of the difficulties related to the use of liquid feed particularly concerns the control of variations of dry matter intake by animals, also pointed out by researchers of the Ministry of Agriculture and Forests (MAF) [30]. It is precisely to overcome these problems that liquid feed, as a direct addition to mixing wagons, is indicated for herds in which the production level is quite homogeneous and in animals with average requirements. These feeds are also indicated to enhance poor basic diets.

## 4. Molasses in Ruminant Nutrition

As has already been mentioned, one of the effects which occur with molasses inclusion in ruminant diets is the increase in dry matter intake due to its superior palatability [31]. This result is even greater when referring to grazing animals (cattle, buffaloes, sheep and goats) if fed with poor quality forage [32]. Under these conditions, molasses has a stimulating effect on the digestive activity of ruminal microbiota, thus improving both the digestibility of coarse quality forage and dry matter intake [33]. In particular, when referring to ruminant nutrition, the effects of molasses on ruminal fermentation merit mentioning, as Andrighetto and Andreoli [34] appropriately specified.

In recent decades, much research has been carried out concerning the effects of molasses inclusion or feed based on molasses on dairy cattle [35,36,37,38,39,40]. However, the results do not always agree since they are often influenced by both the different diets and the different amounts of molasses used.

In fact, molasses efficiency would be strongly penalized (negative results are also obtained) if the limit value of 10% (DM) of the daily ration was exceeded.

In a recent study, de Ondarza et al. [39] analyzed 24 research using a mixed model linear regression analysis. They considered different levels of sugars added in the rations (control, 1.5–3%, 3–5% and 5–7% of DM), days in milk category (within treatment), control milk yield category (within treatment), and several continuous nutrient variables. Results show that, in cows producing >33 kg of milk/d, added dietary sugar had a greater response (2.14 kg of 3.5% FCM/d; *p* < 0.0001) than in cows producing <33 kg of milk (0.77 kg of 3.5% FCM/d). Authors concluded that to optimize 3.5% FCM yield response when feeding additional dietary sugars, a low to moderate starch diet should be fed (22 to 27% of diet DM) in combination with a moderate to high soluble fiber content (6.0 to 8.5% of diet DM). These data assess the high importance of balancing the different CHO fractions since each one of them is able to specifically drive and impact the composition of the rumen microbiome and, more generally, the animal responses.

If moderate doses of molasses are used, this leads to optimization of the ruminal fermentation [4], improved microbial activity, increased protein synthesis, and reduced ammonia content in the ruminal liquid [41]; it follows a reduction in the amount of urea in blood and milk [38]. Greater nitrogen fixation in the rumen leads to a reduced nitrogen loss through the feces and urine, providing beneficial results to the environment, as demonstrated by Hristov [42] and Brito [38]. According to Feroci and Nistri [28], the best way to stimulate protein synthesis by rumen microorganisms is to give proper quantities of molasses and urea in a correct equilibrium with the other dietary components.

In addition, several studies have reported improvements in ruminal butyrate concentrations when dietary sugar levels were increased as a partial replacement for cereal grain starch [4,43,44,45,46], and this mechanism can explain the main sugar advantages observed in the field: (a) higher ruminal efficiency: butyrate is a growth factor for ruminal *epithelium* [47]. Dairy diets, containing higher percentages of sugars, can increase its production and, therefore, promote more efficient energy absorption through the ruminal *papillae*; (b) pH stability: butyrate generates only one H^+^ while other volatile fatty acids (VFAs), such as propionic and acetic acids, generate 2 H^+^. This indicates that, by increasing butyrate proportion production and promoting faster absorption of all VFAs in the rumen, sugars are able to better control rumen pH in comparison with starch [48] and (c) milk fat increase: in vitro studies [49] show that sugars can stimulate the growth of *Butyrivibrio fibrisolvens* bacteria producing butyrate. This leads to an inhibition of the ruminal *trans*−10 biohydrogenation pathway, explaining the higher milk fat synthesis generally observed when substituting parts of starch with sugars in the ration of ruminants.

At this point, it is appropriate to list the advantages which can be obtained in beef and dairy cattle with the use of molasses or liquid feed based on molasses: (a) low system costs and easy storage; (b) the possibility of enriching diets with nonprotein nitrogen, therefore with very low costs and higher feed efficiency; (c) the possibility of adding other byproducts that are less desirable than molasses, and gaining in both digestibility and in dry matter intake; (d) reduction in foot pathology and hypofertility owing to the higher biotin content (vitamin H, Table 5); (e) reduction in the risk of mycotoxicosis and (f) decreased urea content in milk and increased true protein production (the amount of casein in particular). Especially for beef cattle, molasses and, more generally, liquid feed based on molasses, enhance: (a) the attractiveness of these diets during an adaptation period; (b) the production performance (average daily gain and feed conversion rate) [50]; (c) no noteworthy change as regards the carcass quality (color, marbling, tenderness of meat) [34,51].

As for dairy cows, during the dry period, molasses dietary supplementation can be useful for increasing the ingestion and digestibility of coarse fodder and keep rumen glycolytic microflora active, useful in the postpartum period. Referring to the lactation period, particularly to the first weeks, the priority requirement is to avoid the risk of metabolism anomalies, such as ketosis, which is detrimental to productivity, fertility and, more generally, to animal health. Molasses, thanks to its increased palatability and enhanced digestibility, can reduce the gap between diet nutritive supplies and dairy cow needs; it can also be useful through better synchronization between the energetic substrate and nitrogenous degradation [41]. However, more attention must be paid to the diet anion-cation balance, which, in the presence of molasses, could be altered due to higher potassium content, as mentioned above.

According to experts [28], the advantages of molasses in dairy cow feeding can be summarized as follows: (a) increased availability of net energy and digestible protein; (b) increased dry matter intake, milk production, lactation curve persistency and milk fat content; (c) improved fiber digestion and nutritional efficiency; (d) reduction in the risk of subclinical acidosis and rumen sub-acidosis, mucosa damage and innate immune system stimuli, and (e) a reduction in the weight loss of lactating subjects and a decrease in the incidence of ketosis. It is necessary to limit the daily intake of molasses to values below 10% (6.75% of diet DM; [39]). In practice, it is suggested to not exceed 1 kg/head/day for beef cattle and 0.5, 1, 1.5 kg/head/day in heifers and in dry and lactating cows, respectively.

## 5. Molasses in Swine Nutrition

It is evident that the natural function of molasses is to partially replace cereals as an energy source in feed formulations and diets for all categories of pigs. In particular, many researchers have repeatedly pointed out that doses of molasses can be very high, but this requires exact knowledge of the chemical composition, nutritional value and dietary properties of molasses itself and all the other components of the diet. Compared to cereals, the gross energy of molasses is lower, which is related to both the high water and ash contents and the low lipid level (Table 1 and Table 2). Table 2 shows the fatty acid (FA) composition of the lipid fractions of sugar beet and sugar cane molasses; the FA analyses were carried out at the Laboratory of Dipartimento di Scienze Mediche Veterinarie (DIMEVET) of the University of Bologna, using the methods described by Folch et al. [52] and Christie [53]. Another consideration is that carbohydrates in molasses are mainly represented by mono and disaccharides and, therefore, have a lower caloric value with respect to grain polysaccharides (Table 1). The digestible energy of molasses represents only 85–90% of its gross energy; this is due both to the poor exploitation of some undetermined glucose fractions and to the modest presence of nonprotein nitrogenous substances. For this reason, the net energy can be estimated to be approximately 1536–1616 kcal/kg of cane molasses. In practice, as regards the energy supply, it can be estimated that 100 g of maize or barley can be substituted by 140 or 125 g of molasses, respectively [11].

The crude protein content of molasses is low, especially in cane molasses as compared to beet molasses, the latter also being more digestible (Table 1). Given the modest quantity of essential amino acids (the presence of aspartic acid is useful but not essential), the nitrogenous fraction is not very important and is mainly composed of free amino acids and nitrogenous bases. It is, therefore, good practice to balance the amino acid content in diets rich in molasses (Table 4). To this end and to cover the pigs’ requirements, it is necessary to integrate the molasses with peptide sources from protein lysates added directly to the molasses or with amino acids of industrial production. All of these procedures are usually carried out in liquid feed formulations for pigs.

As regards the mineral fraction, in Table 1, similar total ash content is reported between the two types of molasses, and their comparison is extremely interesting if the individual minerals are considered. Beet molasses is richer in sodium, while cane molasses has a higher content of calcium, phosphorus (but not sufficient), magnesium, chlorine and almost all the trace elements (manganese, in particular, Table 3). As regards the potassium content, some clarification is indicated. The high potassium content in molasses, especially in beet molasses, is considered to be a limiting factor for the use of high doses of this byproduct in animal feeding, particularly for pigs. Many nutritionists argue that this element is responsible for a laxative effect if molasses is administered to the animals in large quantities; however, this thesis is more speculative than scientifically documented. In fact, when an excess of molasses is used, it could increase the incidence of renal lesions to a greater extent than the appearance of diarrhea [53]. It has been stated that the effects of potassium-rich cereal diets (in sulfate and carbonate form) do not change the water content of the feces, while they may be responsible for renal dysfunction [54,55].

When considering the sulfur content of molasses, attention had to be paid to the use of molasses in swine feed formulation. As mentioned above, the NRC [16] specified that MTL of sulfur, in an organic or an inorganic form, in non-ruminant feeds is set at 4 g S/kg DM.

Regarding its vitamin content (Table 5), molasses has only a minimal amount of fat-soluble vitamin and, at increased doses of molasses, supplementation with vitamin A, D, and E are, therefore, necessary. On the other hand, molasses is rich in water-soluble vitamin, with particular reference to niacin, pantothenic acid, riboflavin and vitamin H. Biotin deserves detailed discussion; as already stated, more than 90% of biotin is in a bioavailable form in molasses as compared to other raw materials, but often in a low bioavailable form. The role that biotin plays in preventing pig pathologies, in improving the reproductive efficiency parameters of sows, and in the quality of heavy pig fat (reduction of linoleic acid) is now well known [56]. Regarding genetic selection, the new breeding techniques and the systematic use of cereals (wheat, barley, sorghum and bran), which are poor in vitamin H in bioavailable form, are counterbalanced by the biotin content of molasses, which reduces the risk of deficiency phenomena. On the other hand, thiamine is well represented but is almost totally destroyed by the heat treatments carried out during the processing phases.

Regarding the doses of molasses used in heavy pig feeding, some considerations are in order. The type of feed supplied: dry, liquid or semi-liquid, must first be taken into account. Furthermore, the use of molasses in high dietary percentages, together with milk whey, may create some problems which are not easy to resolve. However, after the optimal level has been excluded, when the dietary concentration of molasses is increased, the nutritional efficacy of the feed decreases, and the dose of molasses must be sufficient to provide good zootechnical performance. It has been demonstrated by Christon et al. [57] that the use of molasses in piglets and in growing and fattening pigs can cause decreases in energy and protein utilization efficiency, proportional to the doses administered. This reduction in energy utilization can be justified by both the incomplete digestion of carbohydrates (sucrose), the digestibility of which increases however over time, and by the incomplete intestinal absorption of fructose. The excessive passage rate of amounts of fructose and sucrose into the large intestine could create a hypertonic environment, suitable for the undesired development of microbial fermentation, with consequent production of organic acids which are at the origin of digestive disorders and diarrhea, and therefore not due to the excessive potassium content [54]. As already mentioned above, the worsening of nitrogen retention, which could be associated with the use of molasses, must be compensated for by an increase in dietary nitrogen content by means of an appropriate protein and aminoacidic supplementation. For the aspects described above, it is therefore customary to program an adaptation period (about 10–15 days) before using molasses in large quantities in order to allow the animal digestive system to adapt (specific enzyme induction) and to better use the new feed principles. However, in order to avoid the unpleasant problems described above, it is advisable not to exceed the acceptable dose of 20% of molasses in rations.

According to many researchers [58,59,60], the use of molasses in feed rations for growing and fattening pigs does not change meat quality. The addition of molasses to heavy pig diets, with or without whey, does not have any negative effect on carcass quality [61]. Furthermore, the use of large quantities of molasses in the pig diet can also improve the meat/fat ratio of the carcass by reducing the incidence of fat cuts; however, it can slightly worsen carcass yield. It should be noted that greater caution must be used with (5% of the diet) piglets (up to 20 kg of body weight) and lactating sows; instead, for pregnant sows, boars and fattening pigs, the quantities can also be much higher (10% of the diet) [54]. It is therefore advisable for breeders to use molasses with prudence simply to avoid incurring problems, to reduce the incidence of feed costs and to make pig breeding increasingly competitive. In pig nutrition, the possibility of using liquid feed based on molasses (with a high molasses content; > 60%), which includes adequate lipid quantities by means of specific technologies, considerably increases net dietary energy, making the feed more suitable.

## 6. Molasses for Ensiling Purposes

The use of molasses in the zootechnical field is not limited to the direct administration to animals; however, due to its chemical and fermentation properties, it can be used for ensiling purposes. In fact, its high soluble sugar content allows facilitating the sudden dropping of the pH, thus inhibiting both butyric and proteolytic fermentation to the advantage of lactic fermentation [62]. It is well known that forage legumes, such as alfalfa, clover and even stable grassland, as well as having the lowest simple sugar content, have a high buffer capacity; this is a considerable obstacle to the rapid pH drop required to obtain good quality silage [63]. In these conditions, when the percentage of the legumes reaches a very high level (70–80%), the addition of molasses is very useful [64,65]. Several authors have evaluated the effects of adding molasses at different percentages of inclusion (0 control, 5% and 10%) during ensiling; according to Rameshwar et al. [66], 40 days after silage closing, the silage mass was stabilized with considerably higher levels of lactic acid as compared to the control, and with lower levels of organic acids (acetic, propionic and butyric) and ammonia. The rapid pH lowering inhibits the development of coliform bacteria; this inhibition of proteolytic activity is additionally confirmed by the lower ammonia content at silage opening. This leads one to think that the use of molasses during ensiling improves the organoleptic characteristics of the silage itself and significantly reduces losses due to the conservation time [67]. It should moreover be emphasized that the addition of molasses to excessively wet fodder does not allow such consistent advantages as those described above, and its effectiveness can be affected by losses of nutrients due to percolation [68]. In order to limit these problems, the use of natural clays (Zeolite, Bentonite, Montmorillonite, Clinoptilolite), which have both remarkable absorbing properties towards ammonia and a sequestering activity of mycotoxins, is recommended [1].

The economic convenience of the use of molasses during the ensiling procedure is to be considered strictly tied to the improvement of the organoleptic and nutritional characteristics of the silage. This is, therefore, the case of corn stalks, with low nutritional value and low palatability, or forage legumes, having a low content of easily fermentable carbohydrates. However, the addition of molasses to silage leads to an increase in cost due both to the availability of equipment for molasses distribution and to the slowing down of operating times necessary for ensiling [30]. The addition of molasses (5–10%) to silage also makes it more compact, therefore eliminating air from the silage mass, and increasing both its nutritional value (net energy) and the dry matter content, frequently lacking, especially in grass silage [69]. All these actions increase silage mass by means of both a reduction in respiration and fermentation, and increased acidity and palatability, and, moreover, a decrease in undesired fermentation of molds and *Clostridium* spp. [70]. In addition, the health and welfare status of the animals, the quality of livestock products (milk in particular) and breeder profit will be enhanced.

## 7. Molasses in Urea-Blocks

The use of feed blocks in partial or total substitution of concentrates is a practice in many countries in which grazing is more prominent than confined breeding. Molasses is the main component of a feed block, and, due to its ease of use, it is possible to consistently reduce the incidence of management costs [71,72]. Feed blocks are a solid complementary feed, having a parallelepiped shape, consisting of solid and liquid raw materials with different percentages of inclusion, but, in any case, compacted, with a range of weight from 25 to 750 kg. The blocks have a consistency (hardness) and a structure capable of resisting atmospheric agents so that the organoleptic and nutritive characteristics are not altered. Blocks are feeds which supplement forage, well known as a vehicle for supplying energy, proteins, minerals and vitamin to ruminants, where appropriate and with all limitations provided by law, also drugs [73]. The possibility of being able to modify the inclusion percentages of the individual components is certainly a fundamental characteristic of a block: in this way, it is possible to adapt the feed block to the type of animal (cattle, buffaloes, sheep, goats, horses and also pigs) or to a single production phase.

The feed block is placed in areas easily accessible for animals, raised in group boxes, in a confined barn, and in the wild. It is consumed by animals simply by licking and the ingestion of small amounts (0.5–2.5 kg/head/day for cattle, 0.1–0.7 kg/head/day for sheep); therefore, due to its particular nature, it is fractioned and diluted over time. Each feed block can serve a large group of animals: eight-ten cattle, or twenty to twenty-five sheep and goats; in this way, even the weakest and shyest members of the group will be able to ingest it calmly, thus reducing the problem of hierarchical order, which is created within groups. In relation to pig breeding in the free state (wild, semi-wild, etc.), it is possible that, in the near future, the use of feed blocks will be able to also involve this species significantly with ad hoc formulations.

Since molasses is one of the major components of a feed block, owing to its palatability, a period of adaptation of a few weeks is necessary, especially for grazing animals, before the mineral metabolic profiles of the animals are rebalanced. Due to the high initial intake of the product, in the early stages of adaptation, slight alterations in the metabolic profiles can be found, which, however, return to normality levels in a short time. In addition to making the feed block available to the animals, the adaptation period foresees that the ration will not change. The first dietary change will begin only after the third week, gradually eliminating a proportion of concentrates similar to the amount of feed block used. Once the first period has ended, the feed block, supplied ad libitum, has the characteristic of being self-limiting; therefore, the ration (silage, pasture, unifeed, forage, stalks) is complete and balanced, and the feed block intake will be reduced.

The main function of a feed block is to act at the ruminal level having a simultaneous buffering and stimulating action; thus, fermentation is regularized, and protein and vitamin synthesis is improved, as is the ration feed efficiency (feed conversion rate). The easily fermentable sugars of molasses, associated with urea nitrogen, stimulate rumen microflora activity, allowing the stabilization of the rumen pH, the improvement of fiber digestibility and the tendency of regularizing the digestive functions [72,74,75]. All these benefits result in a) better use of poor-quality forage, b) an increased digestibility of dry matter, fiber, nitrogen and fats [76], c) an increased dry matter intake, especially in high producing cows. The ruminal buffering action carried out by feed block components, thanks to the fractional intake of moderate quantities during the day, allows improving the animals’ health status and consequently reducing the incidence of many common metabolic diseases, such as acidosis, ketosis, meteorism and dislocation. Improvements in digestive and metabolic functions are positively reflected in the productive, reproductive and sanitary efficiency of the herd (reduction in collapses, placenta retentions, metritis, calving-conception interval and the cost of purchasing concentrates, ruminal buffers, yeasts and mineral-vitamin supplements), as well as in increased milk production which is more constant and prolonged over time and of improved quality. It is well-known that the increase in dry matter intake results in increases in feed consumption and in the feed efficiency of the basal diet; therefore, there will be better exploitation of coarse feeds (straw, hay, dry forage, etc., with a consequent cost reduction) as occurs, particularly, in heifers and dry cows. This can lead to both complete body development and to an early start of a productive career in heifers, and a quick bodyweight recovery in dry cows, and the reduction of postpartum pathologies [77].

Very similar considerations can be made for fattening beef cattle; the constant and prolonged intake of urea nitrogen and easily fermented sugars enhance the fiber utilization at the ruminal level. This reflects positively on ingestion and on the consumption of low-quality forage and positively affects daily weight gains and feed conversion rates [73,78].

Briefly summarizing the elements which research and breeding practice have indicated permits affirming that the use of feed blocks, especially in heifers, in dry animals and in transition diets allows: (a) containing production costs and simplifying farm work; (b) increasing feed intake, thus helping to cover nutritional requirements, through the use of relatively less expensive forage; (c) improving the digestive tract functions by enhancing protein and vitamin synthesis, the detoxifying action and fiber digestibility at the ruminal level and (d) maintaining a physiological, metabolic status by reducing the incidence of metabolic disorders with particular reference to acidosis, mastitis, infertility, milk quality and the cheese-making aptitude of milk. All these positive factors translate into the concept of efficiency, health and welfare at a low cost. On the basis of all the considerations made, it is easy to predict a significant expansion of the use of feed blocks in the future, especially in the nutrition of heifers and dry cows, but also of buffaloes and sheep, in extensive productions with low-quality forage. The increased feed efficiency enhances extensive livestock production sustainability and is, therefore, strictly related to climate change [79,80,81]. In fact, the opinion expressed several times by the European Union authorities regarding extensive livestock production will further facilitate the use of feed blocks.

## 8. Conclusions

The aim of the Authors was to summarize the possible applications of molasses in animal nutrition, with the main purpose of enhancing livestock production efficiency. Thanks to its particular properties, this feedstuff can really offer many practical opportunities for meeting the principles of a circular economy, namely improving hay and silage qualities for beef and dairy cattle; managing rumen and intestinal fermentation; enhancing industrial byproduct values with liquid feed, based on molasses, in swine, beef and dairy cattle production and improving extensive livestock production with feed blocks (cows, buffaloes, beef, sheep, goats and pigs).

## Figures and Tables

**Table 1 animals-11-00115-t001:** Composition and nutritive value per kg of sugar beet and sugar cane molasses (modified from the CVB Feed Table 2019. Chemical composition and nutritional values of feedstuffs. www.cvbdiervoeding.nl; and from https://www.feedipedia.org/).

Item	Sugar Beet Molasses		Sugar Cane Molasses		
	CVB Feed Table	Feedipedia	CVB Feed Table		Feedipedia
	(Average)		<475 g/kg	>475 g/kg	
Density, kg/L	1.39		-	1.39	-
DM, g/kg	787	754	724	721	730
Ash, g/kg DM	90	127	112	91	146
Crude protein, g/kg DM	98	142	51	41	55
Crude Fat, g/kg DM	2	2	1	1	1
Nitrogen-free Extract, g/kg DM	597	-	554	582	-
Total sugars, g/kg DM	512	634	454	488	641
Non-starch polysaccharides, g/kg DM	100	-	120	115	-
NPN	59.3	-	61.8	-	-
VEM lactation, unit/kg DM	805	-	601	623	-
VEVI beef, unit/kg DM	890	-	643	669	-
Net energy pigs, kcal/kg DM	1746	-	1536	1616	-

NPN: nonprotein N; VEM: Dutch net energy lactation; VEVI: Dutch net energy beef.

**Table 2 animals-11-00115-t002:** Fatty acid composition of sugar beet and sugar cane molasses lipid fractions.

Fatty Acid (g/100 g FAME)	Sugarbeet Molasses	Sugarcane Molasses
C 8:0	1.66	0.32
C 10:0	1.08	0.03
C 12:0	7.92	0.16
C 14:0	4.77	0.44
C 15:0	0.12	0.29
C 16:0	17.48	24.39
C 16:1 cis9	0.18	0.24
C 17:0	0.20	0.21
C 18:0	10.80	4.56
C 18:1 cis9	22.85	19.96
C 18:1 cis11	0.52	0.91
C 18:2 trans9, 12	0.17	0.15
C 18:2 cis9, 12	29.98	39.20
C 18:3 (n-3)	1.43	7.07
C 21:0	0.12	0.26
C 20:2 (n-6)	--	0.05
C 22:0	0.39	0.47
C 20:3 (n-6)	--	0.06
C 20:3 (n-3)	0.14	--
C 22:2	--	0.46
C 20:5 (n-3)	--	0.07
C 24:0	0.19	0.69
Others FA	7.78	4.04
SFA	44.74	31.83
UFA	55.26	68.17
PUFA	31.55	47.07
MUFA	23.72	21.11
PUFA (n-6)	29.98	39.47
PUFA (n-3)	1.57	7.14
n-6/n-3	19.15	5.53

FA: fatty acid; FAME: fatty acid methyl ester; SFA: saturated fatty acid; UFA: unsaturated fatty acid; PUFA: polyunsaturated fatty acid; MUFA: monounsaturated fatty acid.

**Table 3 animals-11-00115-t003:** Minerals and trace elements contents per kg of sugar beet and sugar cane molasses (modified from the CVB Feed Table 2019. Chemical composition and nutritional values of feedstuffs. www.cvbdiervoeding.nl; and from https://www.feedipedia.org/).

Mineral	Sugar Beet Molasses		Sugar Cane Molasses		
	CVB Feed Table	Feedipedia	CVB Feed Table		Feedipedia
	(Average)		<475 g/kg	>475 g/kg	
Ca, g/kg DM	0.7	1.2	7.9	6.8	9.2
P, g/kg DM	0.5	0.3	0.7	0.6	0.7
Mg, g/kg DM	0.1	0.3	2.7	2.7	4.0
K, g/kg DM	41.0	51.2	41.0	28.8	51.0
Na, g/kg DM	7.2	6.9	1.5	1.0	2.4
Cl, g/kg DM	4.3	-	18.5	21.7	-
S total, g/kg DM	-	5.6	-	-	-
S inorganic, g/kg DM	-	-	8.3	8.2	-
S organic, g/kg DM	0.1	-	0.1	0.1	-
Fe, mg/kg DM	22	154	176	165	173
Mn, mg/kg DM	19	38	24	19	74
Zn, mg/kg DM	13	22	9	12	18
Cu, mg/kg DM	7	17	6	5	6
Mo, mg/kg DM	0.2	-	0.5	0.5	-
I, mg/kg DM	0.3	-	0.9	0.9	-
Co, mg/kg DM	0.6	-	-	-	-

Ca: calcium; P: phosphorus; Mg: magnesium; K: potassium: Na: sodium; Cl: chlorine; S: sulfur; Fe: iron; Mn: manganese; Zn: zinc; Cu: copper; Mo: molybdenum; I: iodine; Co: cobalt.

**Table 4 animals-11-00115-t004:** Amino acid contents per kg of sugar beet and sugar cane molasses (modified from the CVB Feed Table 2019. Chemical composition and nutritional values of feedstuffs. www.cvbdiervoeding.nl; and from https://www.feedipedia.org/).

Amino Acid	Sugar Beet Molasses		Sugar Cane Molasses	
	CVB Feed Table	Feedipedia	CVB Feed Table	Feedipedia
LYS, g/kg DM	0.5	2.1	0.4	0.0
MET, g/kg DM	0.3	0.3	0.4	0.2
CYS, g/kg DM	0.3	1.0	0.6	0.7
THR, g/kg DM	0.7	1.0	1.0	0.7
TRP, g/kg DM	0.2	1.1	0.2	-
ILE, g/kg DM	1.7	0.5	0.9	0.3
ARG, g/kg DM	0.3	1.1	0.2	0.3
PHE, g/kg DM	0.5	0.7	0.5	0.2
HIS, g/kg DM	0.2	0.7	0.2	0.2
LEU, g/kg DM	1.7	3.7	1.1	0.7
TYR, g/kg DM	1.6	3.8	0.4	0.7
VAL, g/kg DM	1.1	2.6	1.9	2.1
ALA, g/kg DM	2.2	2.6	3.4	4.1
ASP, g/kg DM	5.6	7.7	17.8	12.1
GLU, g/kg DM	36.0	67.8	7.6	5.5
GLY, g/kg DM	1.8	2.4	1.4	1.0
PRO, g/kg DM	0.9	1.3	1.0	0.7
SER, g/kg DM	1.7	3.0	1.2	1.0

DM: dry matter; LYS: lysine; MET: methionine; CYS: cysteine; THR: threonine; TR: tryptophan; ILE: isoleucine; ARG: arginine; PHE: phenylalanine; HIS: histidine; LEU: leucine; TYR: tyrosine; VAL: valine; ALA: alanine; ASP: aspartate; GLU: glutamate; GLY: glycine; PRO: proline; SER: serine.

**Table 5 animals-11-00115-t005:** Vitamin content of sugar beet and sugar cane molasses (modified from raw material compendium. 1994. NOVUS).

Vitamin	Sugarbeet Molasses	Sugarcane Molasses
Vitamin A, mg/kg	-	-
Vitamin E, mg/kg	2.4	1.92
Biotin, mg/kg	0.7	0.68
Choline, mg/kg	890	782
Folic acid, mg/kg	0.34	0.12
Niacin, mg/kg	44.6	37
Pantothenic acid, mg/kg	4.58	38.2
Riboflavin, mg/kg	2.2	2.6
Thiamine, mg/kg	1	0.9
Pyridoxine, mg/kg	5.2	4.9

## Data Availability

Not applicable.

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
