# Peer review of "A Review Regarding the Use of Molasses in Animal Nutrition"

_animals, 2021, doi:10.3390/ani11010115_

Round 1

Reviewer 1 Report

Animals 1027641

General remark

Interesting review on the use of molasses in animal nutrition.

The paper refers several times to CVB tables 2018; however a more recent version 2019 with updated data is available and should be used by preference.

Another useful database is Feedipedia (available on line) which would be interesting to mention also in this review as comparison

Specific remarks

Lines 68 : sucrose levels are not correct : in sugarbeet and sugarcan molasses about 50% on DM (see e.g. CVB Tables 2019):

Lines 75-76 : reducing sugars are not fermentable : please give some references (experimental data needed as proof) (Baker 1981?)

Baker, P. (1981) Proc. AFMA Eleventh Ann. Liquid Feed Symp. Amer.

Feed Manufactures Assoc Arlington, VA

Lines 160-238 : mostly related to ruminants : why not incorporate it directly in Section 4; besides liquid feed in pigs is poorly described without any reference. Maybe the “pig issues” can be incorporated in section 5.

Line 330 please refer to the NE value and not to DE values, as stipulated by CVB Tables 2019

Line 209 : please explain fatty liquid feed

Line 262 versus line 311 : some discrepancy : please explain

Lines 344-356 : in the text, the diarrhea issue is mostly related to the K content of the ash fraction; please consider here also the S and more importantly the sulfate content.

Table 3 : only organic S is mentioned, although inorganic S (mostly sulfates) is more important in quantity and also regarding the potential negative effects on animals.

It is important to mention that estimating S from sulfates is simply not correct !

In CVB (2019) tables : sugar cane molasses : 11 g/kg inorganic S and 0.2 g/kg organic S on DM; sugar beet molasses : 0.2 g/kg organic S and no data on inorganic, while Feedipedia reports 5.6 g total S/kg beet molasses.

Regarding S and sulfates in animal nutrition please consider the following in your manuscript (see e.g. EFSA Opinion 2019; Appendix A) : When formulating compound feeds/TMR, consideration should be given to i) some feed materials may already contain, besides S-amino acids, a high background amount of sulfate and/or other S species (e.g. DDGS, biomasses, rapeseed meal, sugarbeet pulp and molasses/vinasses, grass, alfalfa), ii) adverse effects of excess of sulfate/S in animal nutrition are well described (e.g. reduction of feed intake, diarrhoea, disturbances in the well-known Cu x S x Mo interactions in ruminants, negative effects on the absorption of some trace element (e.g. Cu, Se, Zn), polioencephalomalacia), iii) it is impractical to analyse all S species individually in feed. Therefore, the formulation of the feed should carefully take into account the maximum tolerable level of total S, as established by NRC (2005) and set in ruminant diets at 3 g S/kg DM (diet rich in concentrate) and at 5 g S/kg DM (diet rich in roughage) and in non-ruminant diets at 4 g S/kg DM. Also the contribution of S/sulfate present in water for drinking to the total S intake should be considered, especially when the content is high. Kamphues et al. 2014 proposed to consider S-compounds (mainly SO4) as an undesirable feed ingredient and MTL`s should be fixed in the feed law.

Refs :

- EFSA FEEDAP Panel (EFSA Panel on Additives and Products or Substances used in Animal Feed), Bampidis V, Azimonti G, Bastos ML, Christensen H, Dusemund B, Kouba M, Kos Durjava M, Lopez-Alonso M, Lopez Puente S, Marcon F, Mayo B, Pechova A, Petkova M, Sanz Y, Villa RE, Woutersen R, Costa L, Dierick N, Flachowsky G, Glandorf B, Herman L, Karenlampi S, Leng L, Mantovani A, Wallace RJ, Aguilera J, Tarres-Call J and Ramos F, 2019a. Scientific Opinion on the safety of concentrated L-lysine (base), L-lysine monohydrochloride and L-lysine sulfate produced using different strains of Corynebacterium glutamicum for all animal species based on a dossier submitted by FEFANA asbl. EFSA Journal 2019;17(1):5532, 24 pp. https://doi.org/10.2903/j.efsa.2019.5532

- Kamphues J, Dohm A, Zimmermann J and Wolf P, 2014. Sulfur and sulfate contents of feeds – Still or again of interest (in German); Übersichten zur Tierernährung, 42, 81-139.

Therefore please complete the Table 3 with data on inorganic S, organic S and possible sulfates.

Author Response

Reviewer 1

General remark

Interesting review on the use of molasses in animal nutrition.

The paper refers several times to CVB tables 2018; however a more recent version 2019 with updated data is available and should be used by preference.

Another useful database is Feedipedia (available on line) which would be interesting to mention also in this review as comparison

Thank you very much for all the suggestions, I have completed and updated Tables 3 (L190) and 4 (L390) with CVB 2019 and Feedipedia data about sulfur (organic and inorganic).

Specific remarks

Lines 68 : sucrose levels are not correct : in sugarbeet and sugarcan molasses about 50% on DM (see e.g. CVB Tables 2019):

I am sorry for mistake, I have corrected the data of sucrose. (L80-81)

Lines 75-76 : reducing sugars are not fermentable : please give some references (experimental data needed as proof) (Baker 1981?)

Baker, P. (1981) Proc. AFMA Eleventh Ann. Liquid Feed Symp. Amer.

Feed Manufactures Assoc Arlington, VA

I have added the reference (L89 and L590-591)

Lines 160-238 : mostly related to ruminants : why not incorporate it directly in Section 4; besides liquid feed in pigs is poorly described without any reference. Maybe the “pig issues” can be incorporated in section 5.

Liquid feed section concerns both ruminants and monogastrics. In my opinion, it’s limiting to include this section only in pigs paragraph.

Line 330 please refer to the NE value and not to DE values, as stipulated by CVB Tables 2019

As suggested, I have corrected the sentence with the NE values (L377-378)

Line 209 : please explain fatty liquid feed

I have added a sentence explaining how a liquid feed is composed (L233-246)

Line 262 versus line 311 : some discrepancy : please explain

The sentence in L252-264 (old text) has been deleted and rearranged on L 301-311. All the entire period is now referred to a normal/moderate doses of dietary molasses supplementation

Lines 344-356 : in the text, the diarrhea issue is mostly related to the K content of the ash fraction; please consider here also the S and more importantly the sulfate content.

The sentence has been changed (L405-410) I have added the EFSA Opinion 2019 Appendix A: “Sulfates fed at physiological doses to animals are relatively well absorbed. Urinary excretion is the main route.”

Table 3 : only organic S is mentioned, although inorganic S (mostly sulfates) is more important in quantity and also regarding the potential negative effects on animals.

It is important to mention that estimating S from sulfates is simply not correct !

In CVB (2019) tables : sugar cane molasses : 11 g/kg inorganic S and 0.2 g/kg organic S on DM; sugar beet molasses : 0.2 g/kg organic S and no data on inorganic, while Feedipedia reports 5.6 g total S/kg beet molasses.

Regarding S and sulfates in animal nutrition please consider the following in your manuscript (see e.g. EFSA Opinion 2019; Appendix A) : When formulating compound feeds/TMR, consideration should be given to i) some feed materials may already contain, besides S-amino acids, a high background amount of sulfate and/or other S species (e.g. DDGS, biomasses, rapeseed meal, sugarbeet pulp and molasses/vinasses, grass, alfalfa), ii) adverse effects of excess of sulfate/S in animal nutrition are well described (e.g. reduction of feed intake, diarrhoea, disturbances in the well-known Cu x S x Mo interactions in ruminants, negative effects on the absorption of some trace element (e.g. Cu, Se, Zn), polioencephalomalacia), iii) it is impractical to analyse all S species individually in feed. Therefore, the formulation of the feed should carefully take into account the maximum tolerable level of total S, as established by NRC (2005) and set in ruminant diets at 3 g S/kg DM (diet rich in concentrate) and at 5 g S/kg DM (diet rich in roughage) and in non-ruminant diets at 4 g S/kg DM. Also the contribution of S/sulfate present in water for drinking to the total S intake should be considered, especially when the content is high. Kamphues et al. 2014 proposed to consider S-compounds (mainly SO4) as an undesirable feed ingredient and MTL`s should be fixed in the feed law.

Refs :

- EFSA FEEDAP Panel (EFSA Panel on Additives and Products or Substances used in Animal Feed), Bampidis V, Azimonti G, Bastos ML, Christensen H, Dusemund B, Kouba M, Kos Durjava M, Lopez-Alonso M, Lopez Puente S, Marcon F, Mayo B, Pechova A, Petkova M, Sanz Y, Villa RE, Woutersen R, Costa L, Dierick N, Flachowsky G, Glandorf B, Herman L, Karenlampi S, Leng L, Mantovani A, Wallace RJ, Aguilera J, Tarres-Call J and Ramos F, 2019a. Scientific Opinion on the safety of concentrated L-lysine (base), L-lysine monohydrochloride and L-lysine sulfate produced using different strains of Corynebacterium glutamicum for all animal species based on a dossier submitted by FEFANA asbl. EFSA Journal 2019;17(1):5532, 24 pp. https://doi.org/10.2903/j.efsa.2019.5532

- Kamphues J, Dohm A, Zimmermann J and Wolf P, 2014. Sulfur and sulfate contents of feeds – Still or again of interest (in German); Übersichten zur Tierernährung, 42, 81-139.

Therefore please complete the Table 3 with data on inorganic S, organic S and possible sulfates.

Thank you very much for all the suggestions, I have completed and updated Tables 3 (L190) and 4 (L390) with CVB 2019 and Feedipedia data about sulfur (organic and inorganic). Furthermore I have added some specification sentences about sulfur and sulfates (L140-155 and L405-410)

Reviewer 2 Report

On the whole, I think this manuscript is well written.
Are they all right inTable number ?

My specific comments are as follows,
L54-; Table 1. --- Tell us what are NPN, VEM and VEVI in footnote.
What are Net Energy pigs., Metabolisable Energy broilers and Metabolisable Energy rabbits ?
Fermentable Organic Matter r-07 --- What is r-07 ?
Are unit of Coefficients % ?
L91-; Table 4. --- data in Sugar cane molasses <475 g/kg -0.2, -0.1... I cannot understand g/kg DM have negative value.
L93; nl). --- Check font style.
L167; oxygen --- Oxygen
L230-238; It should also be noted ... enhance poor basic diets. --- Is font different ?
L252; D.M. --- DM ? Do you mean dry matter ?
L257-; abnormal increase in butyric acid production --- I have never heard sugars makes only abnormal increase in butyric acid production without increase in acetic and propionic acid. I am saying every time for others sugars increase 'volatile fatty acids' rapidly. Please teach me the mechanism in detail.
L259; the risk of ketosis is high --- When I learned ketosis, abnormal increase in fat mobilization or fed silage with butyric acid are the only reasons. Rather, I thought it was something that should be fed to the ruminants when they were in ketosis to lessen the symptoms.
L264-286; On the other hand, if ... in the ration of ruminants [43]. --- Is font different ?
L269; but not by Broderick [36] and Ghedini [31] --- What does this 'but not' mean ? I'm curious.
L277; Dairy diets --- I am not sure this is OK or daily diets.
L277-L286; When I learned ruminant nutrition, higher percentages of sugars rather increase propionic acid. And wouldn't the milk fat percentage be reduced ?
L311; d) reduction in the risk of rumen sub-acidosis --- I think it could be more dangerous the risk of subclinical acidosis.
L313; decrease in the incidence of ketosis --- Didn't you say the opposite in increasing butyric acid ?
L313; D.M. --- DM
L324; DIMEVET --- What is this ?
L395; with (5%) piglets --- What do you mean (5%) ?
L445-; Blocks are feeds ... law, also drugs [68]. --- Is font different ?

L530; World Journal of Zoology --- OK ?
L531; Journal of Animal Science. --- J. Anim. Sci.
L534; Italian Journal of Animal Science. --- Ital. J. Anim. Sci.
L537; Clinical Biochemistry --- Clin. Biochem.
L547; september --- September
L548; Informatore Zootecnico --- OK ?
L550; Chickens Poultry Science. --- Chickens. Poult. Sci.
L552; november --- November
L554; november --- November
L557; The Professional Animal Scientist --- Prof. Anim. Sci.
L560; Livestock Research for Rural Development --- Livest. Res. Rural Dev.
L568; Zootecnica e Nutrizione Animale --- Zootec. Nutr. Anim.
L579; L’Informatore Agrario --- Inf. Agrar.
L594; The Professional Animal Scientist --- Prof. Anim. Sci.
L620; J. Dairy Sci. --- Italic
L635; Journal of Animal Science and Biotechnology --- J. Anim. Sci. Biotechnol.
L638; Italian Journal of Food Safety --- Ital. J. Food Saf.
L640; Animal Science Journal --- Anim. Sci. J.
L643; J Biol --- J. Biol.
L651; J --- J.
L672; Journal of the Science of Food and Agriculture --- J. Sci. Food Agric.
L676; Queensland Journal of Agricultural Science --- OK ?
L709; Informatore Zootecnico --- OK ?
L716; Agricultural Systems --- Agric. Syst.

Author Response

Reviewer 2

On the whole, I think this manuscript is well written.

Are they all right inTable number ?

I am very sorry for the mistake, all tables numbers were rearranged

My specific comments are as follows,
L54-; Table 1. --- Tell us what are NPN, VEM and VEVI in footnote.
What are Net Energy pigs., Metabolisable Energy broilers and Metabolisable Energy rabbits ?
Fermentable Organic Matter r-07 --- What is r-07 ?
Are unit of Coefficients % ?

The table 1 has been changed, I have added all specifications needed, thanks

L91-; Table 4. --- data in Sugar cane molasses <475 g/kg -0.2, -0.1... I cannot understand g/kg DM have negative value.

Sorry for the mistake, Table 4 (L390) was updated with CVB Feed Table 2019

L93; nl). --- Check font style.

Ok, thanks, the table 4 shift to L381

L167; oxygen --- Oxygen

I have corrected, thanks (L203)

L230-238; It should also be noted ... enhance poor basic diets. --- Is font different ?

I am sorry, I have corrected

L252; D.M. --- DM ? Do you mean dry matter ?

Yes, I have corrected (L300)

L257-; abnormal increase in butyric acid production --- I have never heard sugars makes only abnormal increase in butyric acid production without increase in acetic and propionic acid. I am saying every time for others sugars increase 'volatile fatty acids' rapidly. Please teach me the mechanism
 in detail.

L259; the risk of ketosis is high --- When I learned ketosis, abnormal increase in fat mobilization or fed silage with butyric acid are the only reasons. Rather, I thought it was something that should be fed to the ruminants when they were in ketosis to lessen the symptoms.

The sentence in L252-264 (old text) has been deleted and rearranged on L 301-311. All the entire period is now referred to a normal/moderate doses of dietary molasses supplementation

L264-286; On the other hand, if ... in the ration of ruminants [43]. --- Is font different ?

I am sorry, I have corrected

L269; but not by Broderick [36] and Ghedini [31] --- What does this 'but not' mean ? I'm curious.

In their researches Broderick and Ghedini did not found the reduction of nitrogen loss by the rumen. I have deleted both references

L277; Dairy diets --- I am not sure this is OK or daily diets.

Yes, Dairy is the right word

L277-L286; When I learned ruminant nutrition, higher percentages of sugars rather increase propionic acid.
 And wouldn't the milk fat percentage be reduced ?

To justify my statements I have added some papers:

as regards point b) (Brito, A.F.; Soder, K.J.; Chouinard, P.Y.; Reis, S.F.; Ross, S.; Rubano, M.D.; Casler, M.D. Production performance and milk fatty acid profile in grazing dairy cows offered ground corn or liquid molasses as the sole supplemental nonstructural carbohydrate source. J. Dairy Sci. 2017, 100, 8146–8160. doi: 10.3168/jds.2017-12618)

and for point c) (Sun, X.; Wang, Y.; Chen, B.; Zhao, X. Partially replacing corn starch in a high-concentrate diet with sucrose inhibited the ruminal trans-10 biohydrogenation pathway in vitro by changing populations of specific bacteria. J. Anim. Sci. Biotechnol. 2015, 6, 57).

Particularly this second paper, an in vitro study, in Conclusion section Sun writes: “These results indicate that replacing starch in a high-concentrate diet with sucrose increased butyrate production and inhibited the rumen trans-10 biohydrogenation pathway, which was at least partially due to increased abundance of Butyrivibrio fibrisolvens and decreased abundance of Megasphaera elsdenii.”

L311; d) reduction in the risk of rumen sub-acidosis --- I think it could be more dangerous the risk of subclinical acidosis.

I agree with you, I have added also the risk of subclinical acidosis (L357-358)

L313; decrease in the incidence of ketosis --- Didn't you say the opposite in increasing butyric acid ?

The sentence in L259 has been deleted. and rearranged on L 301-311.

L313; D.M. --- DM

I have corrected (L360)

L324; DIMEVET --- What is this ?

I have corrected (L370-371), it was already written in L5

L395; with (5%) piglets --- What do you mean (5%) ?

5% of the diet, I have detailed it (L454 and L456) and added a reference

L445-; Blocks are feeds ... law, also drugs [68]. --- Is font different ?

You are perfectly right, I have corrected it

L530; World Journal of Zoology --- OK ?

L531; Journal of Animal Science. --- J. Anim. Sci.
L534; Italian Journal of Animal Science. --- Ital. J. Anim. Sci.
L537; Clinical Biochemistry --- Clin. Biochem.
L547; september --- September
L548; Informatore Zootecnico --- OK ?
L550; Chickens Poultry Science. --- Chickens. Poult. Sci.
L552; november --- November
L554; november --- November
L557; The Professional Animal Scientist --- Prof. Anim. Sci.
L560; Livestock Research for Rural Development --- Livest. Res. Rural Dev.
L568; Zootecnica e Nutrizione Animale --- Zootec. Nutr. Anim.
L579; L’Informatore Agrario --- Inf.
Agrar.
L594; The Professional Animal Scientist --- Prof. Anim. Sci.
L620; J. Dairy Sci. --- Italic
L635; Journal of Animal Science and Biotechnology --- J. Anim. Sci. Biotechnol.
L638; Italian Journal of Food Safety --- Ital. J. Food Saf.
L640; Animal Science Journal --- Anim. Sci. J.
L643; J Biol --- J. Biol.

I am sorry but the entire title of the journal is Journal of Biological Chemistry

L651; J --- J.
L672; Journal of the Science of Food and Agriculture --- J. Sci. Food Agric.
L676; Queensland Journal of Agricultural Science --- OK ?
L709; Informatore Zootecnico --- OK ?
L716; Agricultural Systems --- Agric. Syst.

For References paragraph: I have corrected all specifications throughout all this section

Reviewer 3 Report

The review paper “A review regarding the use of molasses in animal” presented for review concerns the possibilities of molasses management in feeding of ruminants and pigs.

The paper is written in detail and thoroughly, but I have a few comments

General  comments

  • The order of tables - Tables appear in a nonchronological order(1, 4 3 2), this requires ordering.
  • Information about sugar cane and beet molasses is described in great detail. In my opinion, however, information about the scale of their production should be added, taking into account the distribution of crops of these plants. More precisely, from this review, we found that molasses is a valuable feed additive/substitute, but there is no information about its resources, so it is not known whether it is worth to deal with it, because the scale of its availability may be limited.
  • The availability of beet and cane molasses is different depending on the region it may be important to use it. It should be mentioned in the text.
  • Most of my comments relate to section “Molasses in ruminant nutrition” which should be changed significantly - comments see in text

In my opinion, the review is suitable for publication after minor revision.

Author Response

Reviewer 3

The review paper “A review regarding the use of molasses in animal” presented for review concerns the possibilities of molasses management in feeding of ruminants and pigs.

The paper is written in detail and thoroughly, but I have a few comments

General  comments

  • The order of tables - Tables appear in a nonchronological order(1, 4 3 2), this requires ordering.

I am very sorry for the mistake, all tables numbers were rearranged

  • Information about sugar cane and beet molasses is described in great detail. In my opinion, however, information about the scale of their production should be added, taking into account the distribution of crops of these plants. More precisely, from this review, we found that molasses is a valuable feed additive/substitute, but there is no information about its resources, so it is not known whether it is worth to deal with it, because the scale of its availability may be limited.

I have added a sentence explaining the actual situation of the sugar world production (L45-54)

  • The availability of beet and cane molasses is different depending on the region it may be important to use it. It should be mentioned in the text

I have added a sentence specifying the situation in Italy (L45-54)

  • Most of my comments relate to section “Molasses in ruminant nutrition” which should be changed significantly - comments see in text

In my opinion, the review is suitable for publication after minor revision.

Comments:

L106-107: This is an important statement.  I am curious about the concentration of biotin in this experience I cannot reach this paper.

I have added sentences adding the biotin amount for dairy cows (L114-124)

L252: I think that this statement is a bit out of place because the rumen ecosystem is very complex.

L253: Only bacteria are listed, and what about other rumen microorganisms - protozoa, which can make up to 50% of rumen microorganisms biomass, .Newbold Charles J., de la Fuente  see Gabriel, Belanche Alejandro, Ramos-Morales Eva, McEwan Neil  The Role of Ciliate Protozoa in the Rumen 

L262-263: It's exactly the opposite see Raul Franzolin,  Burk A. Dehority The role of pH on the survival of rumen protozoa in steers(

L272: To be considered.  You can add information on the effect of molasses on protozoa.  ( uptake of soluble sugars, stabilization of fermentation) see

 1.Constraints To The Efficient Utilization Of Sugarcane And Its Byproducts As Diets For Production Of Large Ruminants By R.A. Leng and T.R. Preston http://www.fao.org/3/s8850e/S8850E26.htm

  1. Accumulation of Reserve Carbohydrate by Rumen Protozoa and Bacteria in Competition for Glucose Bethany L. Denton, Leanne E. Diese, Jeffrey L. Firkins, Timothy J. Hackmann

The sentence in L252-264 (old text) has been deleted and rearranged on L 301-311. All the entire period is now referred to a normal/moderate doses of dietary molasses supplementation

L282: I did not fiund such information in this paper

Yes, all the period (L320-332)

L285-286: You should add thet it was in vitro studies

Ok, I am sorry, I have rephrased the sentence (L329)

Round 2

Reviewer 1 Report

Animals 1027641 revised version

The manuscript has been sufficiently improved.

Only 2 remarks remain :

Lines 137-151 : please refer in this paragraph certainly to EFSA 2019 (Ref 55) as what is written here is a rephrased text originating from a literature study published in an EFSA 2019 opinion; please update/complete therefore this paragraph.

Lines 139-140 : represented …..are presented : please improve this sentence

Author Response

Only 2 remarks remain :

Lines 137-151 : please refer in this paragraph certainly to EFSA 2019 (Ref 55) as what is written here is a rephrased text originating from a literature study published in an EFSA 2019 opinion; please update/complete therefore this paragraph.

I have added a sentence citing the EFSA 2019 opinion, thanks (L143-144 underlined in yellow); the new EFSA opinion reference number is 18 (L622-630)

Lines 139-140 : represented …..are presented : please improve this sentence

I am very sorry, I have corrected the sentence (L144-147 underlined in yellow)

Furthermore, I have added one more reference, that strongly match the topic (L568 and L802-804 underlined in yellow)